# Peer review of "Mitigating adversarial manipulation in LLMs: a prompt-based approach to counter Jailbreak attacks (Prompt-G)"

_PeerJ Computer Science, doi:10.7717/peerj-cs.2374_

## Round 0.1 · original submission · Minor Revisions

The reviews indicate that the manuscript makes an interesting contribution to the state of the art, with a promising approach to preventing jailbreaks in LLMs through the Prompt-G framework. However, some aspects that need clarification and improvement are highlighted.

Strengths:
- The research is relevant and well aligned with current needs in the LLM field, offering new insights into a relatively unexplored topic.
- The structure of the article is logical and follows the standard academic format, with well-defined sections.
- Reviewers appreciate the innovative approach and the availability of the source code, which contributes to the reproducibility of the study.

Improvement suggestions:
1. Structure and clarity:
- It is suggested to integrate Section 2 into the Introduction as a subsection, with the addition of bullet points summarizing the main contributions of the article. Furthermore, it would be useful to improve the clarity and conciseness of the Abstract and Introduction.
- Some figures could be better labeled and described to ensure they are understandable on their own.

2. Experimental design:
- Reviewers recommend clarifying the choice of metrics used, including standard metrics such as precision, recall, accuracy and F1-score, given that the model is treated as a binary classifier.
- You need to provide more details on the methodology of calculating the attack success rate and justify the choice of threshold used for binary classification in Section 4.5.
- The article should discuss the distribution of toxicity scores, beyond simple binary classification.

3. Validity of the results:
- Results should be presented with greater emphasis on statistical significance and, if possible, compared to baseline models for a more in-depth evaluation.
- Further discussion regarding the scalability of the Prompt-G framework and its applicability in practical scenarios would be of great value.

Conclusion:
The reviews suggest that the manuscript, while promising, requires minor revisions to improve clarity, coherence, and methodological robustness. Implementing the suggestions given by the reviewers will help strengthen the impact of the study.

·

Basic reporting

The article offers an interesting increment of the state of the art.
There are a few comments to note:
- Section 2 should be a subsection of the introduction and not a separate section
- It would be helpful to add, before section 2.1, a couple of bullet points summarizing what the contributions of the paper are.

Experimental design

- Table 3 shows a range of data showing how--in fact--the LLM was treated as a binary classifier. If so, why were standard metrics (confusion matrix, precision, recall, accuracy, F1-score) not used?
- It is unclear how the attack success rate was calculated: for example, in SQLI cases, was it verified that the output matched a real SQLI? The LLM could produce non-malicious queries and pass them off as malicious
Section 4.5 describes the toxicity analyzer as providing a toxicity score and a binary classification. Where does the binary classification originate? Is there a predetermined threshold? If so, justify the choice of threshold value.
- Why was the study focused solely on the binary response? If you have a toxicity score on a normalized scale, evaluating the distribution of those scores is even more interesting.
- "Similarity" is mentioned many times in the text, but I cannot find a mathematical formulation of it in the text or a clear reference to the metric used

Validity of the findings

The study is interesting, but the results shown, in light of my previous comments, use metrics that could be standardized. In need of clarification is the issue related to attack assessment. The simplification to a binary classification problem, moreover, hides additional possible results that could greatly enhance the value of the study.
While the paper is promising, evaluations on the merits should be deferred to a later updated version.

Reviewer 2 ·

Basic reporting

In general, the language is professional and easy to understand. For improved clarity, there are a few sporadic grammatical and phrasing errors, though. To guarantee that the reader understands the main points right away, the abstract and introduction, for example, could be clearer and more succinct.
The paper gives a fair overview of the history of LLMs and jailbreak attacks with pertinent citations.
The paper's introduction, methodology, results, and discussion sections are logically separated, adhering to the standard academic format. The paper makes sense from beginning to end, but to improve coherence, some sections—like the "Goal Conflict" discussion—might be better incorporated into the primary narrative.
Although they could be more accurately labelled and described, figures are pertinent and help to understand the concepts covered. Ensure each figure is of excellent quality and that the captions give enough information for the figures to stand alone.

Experimental design

This study tackles a pressing problem in the field of LLMs by concentrating on the comparatively unexplored field of prompt-based prevention of jailbreak attacks. The study offers fresh insights and falls within the journal's purview, especially its suggested Prompt-G framework.
More explicit descriptions of the datasets used, including their source, nature, and preprocessing methods, should be provided, as the methodology is explained in sufficient detail to grasp the approach. To increase the study's rigour, the metrics selected and their rationale might also be improved. It might be useful to add a mathematical formulation of such metrics.
The authors mention the availability of source code, which is a positive step toward reproducibility. It would be beneficial to include more details on the environment setup and dependencies to facilitate other researchers' replication of the study.

Validity of the findings

Although the data analysis looks solid, there is room for improvement in the presentation of the results, especially concerning the statistical significance of the conclusions. The results would be more insightful if the methods used to measure the success rates of preventing jailbreak attacks were made clear and compared to baseline models.
The data support the conclusions reached, but they could be examined more closely to address potential drawbacks and suggest directions for further study. It would be beneficial to talk about Prompt-G's scalability and its use in practical situations, for instance.

Additional comments

Several technical terms and ideas, like embedding techniques and vector databases, are introduced without enough context.
The conversation about future research could be extended to cover possible restrictions on the current study, like the applicability of the results to other kinds of LLMs or the resilience of the suggested framework in various attack situations.

---

## Round 0.2 · accepted · Accept

The document has been accepted as the authors have skillfully followed all the recommendations from the reviewers and successfully addressed all the requested revisions.